# Effect of Animal-Assisted Therapy on Patients in the Department of Long-Term Care: A Pilot Study

**DOI:** 10.3390/ijerph16081362

**Published:** 2019-04-16

**Authors:** Kristýna Machová, Radka Procházková, Petra Eretová, Ivona Svobodová, Ilja Kotík

**Affiliations:** 1Department of Ethology and companion animal science, Faculty of Agrobiology, Food and Natural Resources Czech University of Life Sciences, 16500 Prague, Czech Republic; eretova@af.czu.cz (P.E.); svobodovai@af.czu.cz (I.S.); 2Department of Statistics, Faculty of Economics and Management, Czech University of Life Sciences, 16500 Prague, Czech Republic; prochazkova@pef.czu.cz; 3Department of Long Term Care, Faculty of Medicine, Charles University, Central Military Hospital—Military University Hospital in Prague, 16902 Prague, Czech Republic; ilja.kotik@uvn.cz

**Keywords:** animal-assisted therapy, long-term care, elderly, well-being, dog

## Abstract

Long-term hospital stays might have a negative psychosocial impact on our patients. One way to positively activate hospitalized patients is to introduce animal-assisted therapy (AAT). A total of 72 individuals participated in this research. The experimental group comprised 33 patients (8 males, 25 females), while the control group contained 39 patients (11 men, 28 women). The participants in the control group were aged from 58 to 100 years and the experimental group featured participants aged from 51 to 95, for whom AAT was included alongside standard care. Blood pressure, heart rate, Barthel index, and general mood were measured in both groups. Results did not reveal any changes in blood pressure, heart rate, or Barthel index in comparison between groups. A great influence was noted in assessment of the mood of the patients. The inclusion of AAT did not affect physiological parameters, but it exerted a significant effect on the psychological well-being of the patients.

## 1. Introduction

Hospitalized patients, especially the elderly, are at risk of “hospitalization syndrome”, a condition associated with staying long-term in hospital. Symptoms comprise of gradual loss of orientation and cognition, loss of patient’s interest in their surroundings, and unwillingness to participate in group therapy or maintain contact with others [1]. Typically, activities that had provided joy in the past are discontinued, the individual wishes to stay in bed and prefers to spend the entire day in their room or sleeps. Indeed, a person becomes increasingly lethargic until they lose all interest in matters. This syndrome has many causes, the most common being medication, infection, response to surgery, isolation, and dehydration [2]. Another factor that can contribute to the development of hospitalization syndrome is the routine and stereotypical course of the days [3].

Consequences can be both psychological and physical in nature, and a permanent change in cognitive function may occur. Nevertheless, sufficient appropriate external stimuli could prevent against its onset. However, it is crucial that any such motivations are of interest to the patient. During the process of rehabilitation, frequent family visits and patient support services are particularly important. The presence of an animal supplements these strategies, since a large number of hospitalized patients have pets, and contact with the animal reminds them of home and the stimuli they received in their own environment [4].

Social and healthcare facilities already run a number of activity programs, such as music or art therapy, cooking, cognitive training, and memory training. It would be possible to introduce an animal into such activities or separately, whether they are put on for a group or an individual. That kind of initiative might prove to be of assistance to those with pets of their own, effectively acting as a bridge between their lives outside the establishment and their present circumstances. Indeed, the efficacy of other therapies may be increased if the patient is motivated and interested in the participation and the content of the therapy [5]. In fact, studies have confirmed that some individuals who had previously refused to join in group therapy actually came along to sessions if they knew a dog would be present. Additionally, the duration of treatment was longer in sessions with the dog in attendance [6].

Furthermore, hospital settings often lack opportunities to engage in meaningful communication, which is an essential part of human life, as well as crucial for motivation and the desire to make progress. A dog can constitute a good mediator of communication, acting as a catalyst for building relationships and providing a common topic for conversation [7]. In this context, people gain a sense of inner calmness and contentment, in addition to being better able to manage social interactions and take in information from the environment [8].

Rodin and Langer [9] stated that an option for supporting long-term hospitalized patients would allow the patients to decide how they wished to spend their time, which would consequently make them feel healthier and better than they would otherwise. The importance of this research lies in the fact that such patients would be encouraged to take small steps to make their own choices and personal commitments. Animal assisted therapy (AAT) might then act as a supportive measure, permitting the patient to opt for the therapies they wish to partake in, a distinct benefit over receiving physiotherapy.

The aim of this study was to determine whether AAT has a positive effect on patients in long-term care and whether these treatments affect parameters such as blood pressure, heart rate, and Barthel index. To this end, the hypothesis was that a positive influence would be exerted, pertaining to the general mood and Barthel index of the patients, thanks to the dog helping to initiate and mediate pleasant experiences for the long-term care patients. We expected the blood pressure and heart rate to be the same in both groups; that the values would remain at the physiological level; and that therapy would not lead to any pathological increase or decrease.

## 2. Methods

### 2.1. Participants

A total of 72 patients participated in this study. The experimental group comprised 33 patients (8 males, 25 females), whereas the control group was made up of 39 patients (11 men, 28 women). The participants in the experimental group were aged between 51 and 95 years (84.5 ± 12, mean ± standard deviation); the control group contained persons between 58 and 100 years of age (87 ± 10.2, mean ± standard deviation). The experimental cohort received rehabilitation supplemented with the assistance of the dog, while the control group was given rehabilitation treatment in a standard way.

This study involved inpatients that were expected to be housed in a long-term care unit for at least three months. The diseases that affected patients were stroke, mild dementia or mild cognitive disease, or cancer. However, the goal was not to overcome the symptoms of such diseases, but to enhance the general condition and well-being of the patients. For this reason, the specific illness was not taken into account as a negative factor for the overall study. Research was conducted under conditions corresponding to real practice. Patients displaying happiness during an encounter with the dog, and who agreed to engage with him in therapy, were deemed suitable candidates. The cohort of patients was randomly assigned into either the control or experimental groups, utilizing envelopes for such randomization. The project was approved by the Ethics Committee of the Central Military Hospital in Prague (Military University Hospital) and evaluated as a non-invasive observational study. It was also approved by the Ethics Committee of the Czech University of Life Sciences in Prague. The study and its methodological procedure adhered with the requirements of European Union and Czech legislation (Act no. 246/1992 coll. on animal protection as amended by Act no. 162/1993 coll.).

### 2.2. Measurements

Diagnosis, rehabilitation, and testing were carried out at the Department of Long-term Care. Data collection and inclusion of patients in the study were carried out shortly after the patients had been admitted to hospital for long-term care.

Mood was evaluated by a Likert scale that allowed patients to rate their general mood on a scale of 1 to 10 [10,11] on every day of therapy. The people in the experimental and control groups were asked by the same therapist to evaluate how they felt. The instruction received by patients was as follows: rate the mood you have right now on a scale of 1 to 10, with 1 being the worst mood and 10 being the best mood. The 10-point scale was chosen in order to improve patients’ understanding compared to a 5-point scale, since patients may have otherwise been confused with using a 5-point scale due to grading at school, where 1 is the best classification. Those in the experimental group were interviewed once a week before and after therapy sessions. For the control group, two measurements were reported from the same day in the morning and after a one-hour interval once a week as well (on the same days as patients in the experimental group) at times they were involved in the normal regime of the hospital.

Methods for evaluating physiological functions and scoring the Barthel index were based on requirements demanded by the nursing documentation; both were carried out at the commencement of the study, repeatedly over the following three months (every three days) and at the end of the test period. Blood pressure and heart rate were measured using pressure gauge. The Barthel index, which assesses self-sufficiency, was incorporated and gauged within the nursing documentation [12,13].

The patients were all housed in one department. Participation in the study was voluntary, and any individual could quit at any time, although no one chose to do so. In addition, everyone gave their permission for the results of the study to be published and for photographs taken during the course of their therapy to be used.

### 2.3. The Therapy Dog

The therapy of the experimental group was conducted with the participation of a five-year-old border collie named Mia. Prior to her joining the therapy sessions, Mia undertook AAT certification. Her training was focused on obedience, even under special circumstances, on her ability to cope with stressful situations, and on her handler’s ability to gauge her behavior during unusual situations. The characteristic feature of Mia is her constant interest in people and working with them. She has no problem being close to strangers, or playing or interacting with them. Notably, Mia shows no aggression towards humans or animals.

The long-term inclusion of Mia in AAT is possible due to careful monitoring of her welfare and respect for her needs. Herein, the content of her “work day” varied, thereby preventing any sense of burnout. She was always given plenty of time to rest, had permanent access to water and she only received kibbles in limited quantities as treats to avoid any overfeeding.

### 2.4. General Procedures

All patients received standard nursing care, therapy, and patient activation under the auspices of rehabilitation nursing. The sessions for members of the experimental group were attended by the therapy dog Mia. Her visits took place once a week over a period of 12 weeks, and the individuals in question also came across her at random either in the department or in group activities. Each individual therapeutic session lasted approximately twenty minutes. Prior to every such visitation, the therapist consulted the condition of the patient with a physician, occupational therapist, physiotherapist, or the nursing staff, and together they evaluated the objective of the upcoming therapeutic session. The aims of sessions were cognitive function training, relaxation, basal stimulation, gross or fine motor skills training, co-operation within the treatment regimen, and prevention of hospitalization syndrome. The therapies selected were based on the needs of the given patient and on his or her current state of mind. Mia accompanied clients on outdoor walks, played fetch with a ball, or did short obedience exercises. She was able to interact positively with patients and generate conversations just through her presence.

### 2.5. Statistical Analysis

The Statistica 13.2 program (Informer Technologies, Inc., Palo Alto, CA, USA) was used to evaluate the data obtained. Initially, exploratory data analysis was performed on the file to verify the independence of sampling, as well as homogeneity of the samples and the normality of their distribution. After carrying out the Shapiro–Wilk test, the data were evaluated via histograms and standardized probability charts.

Pearson’s chi-squared test was employed to test variances in the subjective perception of the patients regarding their mood before and after therapy, as well as to generalize the proof of variance. The strength of dependence was judged on the basis of Pearson’s contingency coefficient (C). The coefficient of association was utilized to assess strength of association. Regarding the four-way contingency table, the chi-squared test confirmed the statistically significant dependence of the qualitative parameters under evaluation. Subsequently, odds ratios were acquired and interpreted, allowing for a relationship to be discerned between the associated frequencies in the four-way table.

For measuring the heart rate and blood pressure, the baseline (the input of the study) was selected as the median of the first five measurements, while the final value (upon completion of the therapeutic cycle) equaled the median of the last five measurements.

The frequencies of the measured data proved different for the individual patients. The data analyzed mostly showed a normal Gaussian distribution. In order to test differences in measured values at the beginning and end of the therapy, a paired-samples *t*-test for frequency-dependent selections was applied. A two-sample *t*-test was employed to assess variances in the results of the experimental and control groups.

A more detailed assessment was conducted for the Barthel index, which showed the greatest variability and improvement. In order to assess the period prior to the first signs of improvement in the experimental and control groups on the Barthel index, the Shapiro–Wilk test was applied to verify the normality of distribution, and the differences were then checked by a two-sample *t*-test for independent samples.

## 3. Results

### 3.1. Patient-Reported General Mood

A statistically significant relationship was seen between the success rate for scores on the Likert scale and the chosen form of therapy. The experimental group that interacted with the dog showed more positive changes in general mood than the control group. When testing the improvement in mood after therapy, the value of the test criterion was chi-squared test = 269.9182; df = 6; *p* < 0.001. A statistically significant difference in post-treatment feelings in the control and experimental groups was demonstrated with 95% confidence. The dependence force measured by the contingency coefficient (C = 0.4442) can be marked as moderate. Comparing the shifts between mood before and after sessions in frequency revealed that the greatest difference between the control and the experimental groups pertained to no shift in the parameter of mood whatsoever, which occurred 2.5 times more frequently for the control group than the experimental group. In fact, there was even a negative shift in the control group that was not recorded for the experimental cohort. Notably, the experimental group demonstrated a positive two-point shift in said parameter twice as often as the control group. A comparable one-point shift for improvement in general mood was discerned for both groups.

The results for the subjective well-being of the control group remained more or less unchanged from the first and second evaluation of mood in the morning of the same day. The experimental group displayed variation between the level of mood recorded in the morning and then following therapy that day with the dog. In this group, there was also a slight shift between the initial rating of mood and the rating of mood after the 15-week cycle of therapy. The experimental group demonstrated a statistically significant difference in shift in mood prior to and following a therapy session with the dog (the value of the Wilcoxon pair test criterion W = 22; df = 22; *p* < 0.001). However, variance between satisfaction in the morning and the second measurement was also seen in the control group (the value of the Wilcoxon pair test criterion W = 47.5; df = 24; *p* < 0.005). Complete results are shown in Figure 1.
Start in the morning—median of values reported by patients at the commencement of the 15-week cycle of AAT from three individual discussions prior to purely standard therapy.Start in the evening—median of three values reported by patients at the commencement of the 15-week cycle of AAT after three such therapy sessions.Finish in the morning—median of patient values at the end of the 15-week cycle of AAT, from three discussions in the morning before a specific dog therapy session prior to the end of the research cycle.Finish in the evening—median of values reported by patients at the end of the 15-week cycle of AAT, from three individual discussions after a specific dog therapy session prior to the end of the research cycle.

A statistically significant difference (*p* = 0.00005) was observed between a comparison of results for the experimental and control groups upon completion of the therapeutic/research cycle, with the patients of the experimental group feeling significantly better than those of the control group (who did not experience the prescribed AAT) (Figure 2).

### 3.2. Heart Rate and Blood Pressure

No statistically significant variances between the experimental and control groups were evident for heart rate, whether in comparison of initial and output values or in comparison of the groups. Additionally, no statistically significant variances were observed in values for blood pressure. Neither intra-group or inter-group (between experimental and control group) differences were observed between the input and output values. Table 1 presents the summary statistics of before and after animal-assisted therapy in parameters of systolic and diastolic blood pressure, pulse, and Barthel index. Table 2 shows results of differentiation testing in groups before and after therapy. Results of testing differences between groups is provided in Table 3. The values of both groups remained within the physiological range over the entire follow-up period.

### 3.3. The Barthel Index

As described for the two parameters above, input and output values were tested for the Barthel index, and no statistically significant differences between values for the experimental and the control groups were observed, either within the experimental group or between the individual groups. No significant changes in Barthel index values were observed in either of the groups. Patients in both groups did not improve or get worse.

## 4. Discussion

The aim of this study was to evaluate the influence of a therapy dog on the health status of patients accommodated in the long-term care department at a hospital. This study also evaluated parameters that formed part of standard nursing documentation, building up a general picture of the health of the patient. Said parameters comprised assessment of heart rate, blood pressure, the Barthel index, and general mood.

No difference was discerned between the experimental and the control groups in heart rate and blood pressure. This study did not reveal any tendency for variation in these parameters. Odendaal and Meintjes [14], Cole et al. [15], Allen et al. [16], Allen et al. [17], and Allen [18] unequivocally demonstrated that interacting with animals resulted in lower blood pressure and a reduced heart rate. Indeed, Odendaal and Meintjes [14] and Cole et al. [15] highlighted such a change of this nature after merely a few minutes of such interaction. In a study by Allen [18], the participants of the study adopted the animals, so long-term measurements were involved. Allen et al. [17] reported that blood pressure was measured as part of the therapy, later demonstrating that it was also reduced. Said study, however, involved dogs being assigned to a number of households, whose members remained in daily contact with the animals. All the studies were working with patients or participants with pathological levels of blood pressure, cardiovascular disease, or who were under acute stress. From the perspective of changes in blood pressure after treatment accompanied by animals, the results reported therein are rather unclear. Therefore, reduction in blood pressure would not appear to be completely evident, although it seems that claims made in support of it have been commonplace. The question remains as to whether the daily presence of a dog would actually diminish the effect of said presence, should the dog become part of daily life at a hospital. Any further study would benefit from optimal frequency of AAT and appropriate measurements, including gauging long-term effects.

Even in relation to ADL (activities of daily living, as assessed by the Barthel Index), no statistically significant shift was discerned in either group. Indeed, no statistically significant differences in input and output values were observed between the control or experimental groups. However, it is important to consider that these were individuals at around the age of 90 years; since these patients were hospitalized with chronic illnesses, it is understandable that any significant shift in their ADL would be exceptional.

Importantly, though, the patients seemed to have benefited from the visits by the dog through signs of improvement in general mood. The patients would anticipate seeing the dog in the department. Said people would link the regular visits by the dog to their room with particular days; such visits lent some kind of meaning to their otherwise drawn-out daily routine. Their reactions to the presence of the dog certainly were not neutral. Such emotions are revealed, as Francis and Pennebaker [19] put it, and it could appear that the animal helped the patients experience such high emotions or let them go to the surface. The presence of the animal additionally formed a subject of conversation amongst fellow patients, in the room or elsewhere, and the attending staff, as well as with members of the patients’ families. Furthermore, elders frequently do not wish to complain or bother anyone, but they can express their feelings to a dog. Similar studies on autism in children have also revealed that sufferers display a sense of contentment and smile more [7,8].

Therefore, it could be hypothesized that the presence of a dog allows other thoughts to enter the minds of the patients. Herein, patients put aside worries about their health, instead starting conversations on ordinary subjects or choosing to remain quiet [20]. Some patients simply sat quietly when petting the dog, while others chatted away when stroking it. The possibility of direct physical contact granted them the pleasant sensation that they could relax and calm down [21].

Another important aspect observed was the readiness by the patient to see the animal. Often the patients had family members or friends bring snacks for it and the dog was the common subject for the visitors and the patients. Interestingly, patients frequently uttered the words “I can finally be useful”. This finding was consistent with a study by Clark et al. [5], which also highlighted the importance in therapy of inherent interest and a sense of meaning as a person.

Some studies describe that the presence of an animal distracts the attention of participants from pain and from thinking about problems [22,23,24,25]; instead, it directs the thoughts of the patient towards the dog and the activities associated with the therapy in question. Naturally, the option exists to move such activities to outdoor areas, beyond the confinements of the healing facility, giving the patients the opportunity to stay outside in the sun and in the company of others, perhaps after spending a few months inside, to choose to dress themselves, do their hair, and to walk the dog, thus exerting a positive influence on their well-being [6,26].

### Limitations

Several questions remain unanswered. One of the limits of this study may be the variety of illnesses suffered by the patients and the different lengths of time that had passed since their initial hospitalization. Another potential limit is the factor of the given therapist or dog. Indeed, the study was conducted with the participation of just one therapist and one dog, hence the personality of the former or the character and temperament of the latter may have impinged upon the outcome of therapy. Lastly, the medication the individual received could have affected the blood pressure measured; this is important with respect to chronically ill patients such as these, since their blood pressure is managed to keep it as stable as possible. Nevertheless, the positive aspect of this study was the fact that research did not take place under laboratory conditions but in a real environment.

## 5. Conclusions

Incorporating AAT into the rehabilitation process for long-term inpatients did not cause any shift in their physiological function or bring improvements in ADL, but patients did significantly improve from a psychological viewpoint. Since these were chronically ill individuals at risk of hospitalization syndrome, any such support mechanism is crucial to their well-being. Should improvement in general mood be the only benefit resulting from AAT then it is still important, as it is necessary to allow elderly and/or long-term care patients to experience dignity and positive moods.

## Figures and Tables

**Figure 1 ijerph-16-01362-f001:**
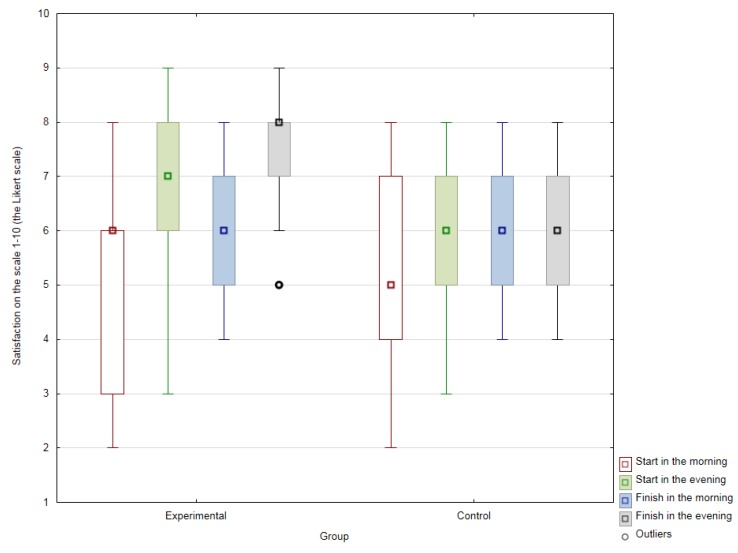
Comprehensive comparison of values reported by patients for subjectively perceived general mood on the scale 1–10 (the Likert scale). Figure terminology is described below.

**Figure 2 ijerph-16-01362-f002:**
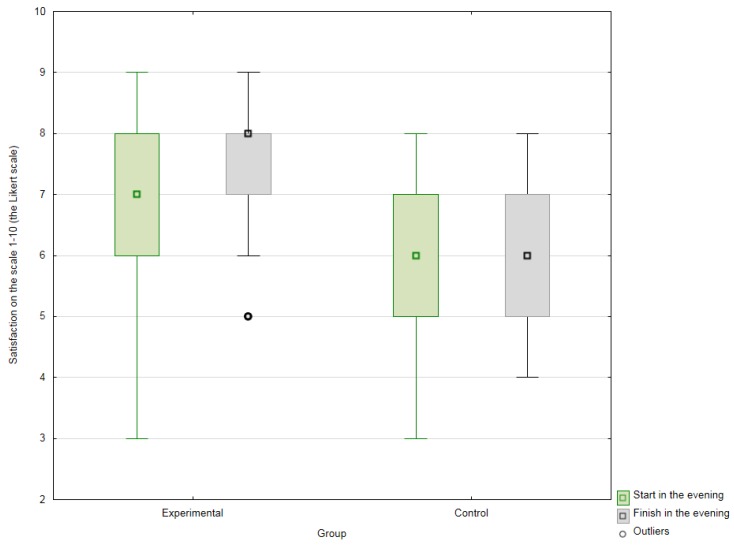
Differences between values at the end of therapy sessions with the dog, at the start of the 15-week cycle of therapy and the end, for the experimental group; values for the control group at hourly intervals from the morning, at the start and end of the 15-week research cycle.

**Table 1 ijerph-16-01362-t001:** Descriptive statistics before and after animal-assisted therapy.

Group	Observed Parameter	Before Therapy					After Therapy				
		Mean	SD	Min	Max	Median	Mean	SD	Min	Max	Median
Control group	Blood pressure systolic	131.4	15.3	91	160	130	129.95	20.2	77	175	132
	Blood pressure diastolic	78.4	13.3	54	144	79	75.7	7.4	47	94	76
	Heart rate	74.3	7.8	58	100	73	72	10.6	52	94	72
	Barthel index	44.9	25.2	0	95	40	44.5	28.6	0	95	45
Experimental group	Blood pressure systolic.	136	19.4	100	171	137	128.2	22	77	166	128
	Blood pressure diastolic	80.2	8.45	62	99	82	78.2	13.5	47	124	78
	Heart rate	76	9.8	51	95	76	75.7	12.4	48	104	73
	Barthel index	50.9	25	5	95	55	50.3	27.2	5	95	55

**Table 2 ijerph-16-01362-t002:** Results of differentiation testing in groups before and after therapy (paired-samples *t*-tests were used for dependent selections).

Observed Parameter	Control Group			Experimental Group		
	*t*	df	*p*-Value	*t*	df	*p*-Value
Blood pressure systolic	0.5092	38	0.6135	1.6811	32	0.1025
Blood pressure diastolic	1.3563	38	0.1847	0.7273	32	0.4723
Heart rate	1.2319	38	0.2255	0.1454	32	0.8853
Barthel index	0.1197	38	0.9053	0.2006	32	0.8423

**Table 3 ijerph-16-01362-t003:** Results of testing differences between groups (*t*-tests were used for independent samples).

Observed Parameter	Measurement	Average Diameter Testing			Scattering Testing		
		*t*	df	*p*-Value	*F*	df	*p*-Value
Blood pressure systolic	B	1.1142	70	0.2690	1.6063	70	0.1616
	E	−0.3549	70	0.7238	1.1873	70	0.6080
Blood pressure diastolic	B	0.6476	70	0.5194	2.4897	70	0.0098
	E	1.0025	70	0.3196	3.3648	70	0.0004
Heart rate	B	0.8120	70	0.4195	1.5863	70	0.1729
	E	1.3266	70	0.1889	1.3805	70	0.3393
Barthel index	B	1.0044	70	0.3187	1.0112	70	0.9817
	E	0.8801	70	0.3818	1.1077	70	0.7725

B—At the beginning of measurement; E—At the end of measurement.

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
