# Peer review of "Effect of Animal-Assisted Therapy on Patients in the Department of Long-Term Care: A Pilot Study"

_ijerph, 2019, doi:10.3390/ijerph16081362_

Round 1
Reviewer 1 Report
Brief summary
This paper presents a pilot study looking into the effects of animal-assisted therapy on hospitalized patients in long-term care. The authors compared the outcomes of patients receiving standard therapy care and those of patients whose therapy was supplemented with the assistance of a therapy dog. The inclusion of a control group was a strength of this study. Specific outcome measures included heart rate, blood pressure, the Barthel index, and mood. The authors found an effect of the therapy dog on patients’ moods.
Broad comments
A strength of this pilot study was the inclusion of a control group – not all studies on animal assisted therapy (ATT) include this important component. Moreover, the authors included several physiological measurements and one psychological measurement in their investigation. Even though there were no differences found between the two groups regarding the physiological factors, these null findings are nonetheless important to the literature on ATT which perhaps has suffered (and to some extent, because of studies with insufficient controls) from an overestimation of the potential benefits of ATT. This study also adds to the literature which has found animals a positive influence on people’s psychological functioning and addresses an important population - hospitalized patients. Knowledge gained as a result of this study has the real potential of positively impacting the lives of patients. Overall, each section would be more clear with improvements in the wording and phrasing. A particular example (others are also highlighted below) is the discussion of how mood was assessed.
Specific comments
1. A main study variable was patient mood or ‘sense of satisfaction’ (line 76) but with the current wording it is unclear exactly how this was measured. First, with the current wording “A mood was evaluated by the Likert scale to assess the sense of satisfaction of the patient” (line 76) it is unclear if the authors were trying to asses patients’ general moods or their sense of satisfaction - or if these are considered the same construct or different. The rest of the sentence (lines 76-77)…”which made it possible to ascertain their current attitude and the approximate degree of the same” is one example of wording whose meaning is impossible to decipher. Most importantly, it is unclear how exactly patients’ sense of satisfaction was actually evaluated – if they rated their satisfaction on a scale of 1-10, or if this rating was judged based on a more general answer, and if so by whom? The authors do write on line 80: “Their answers were judged on a scale of one to ten”. But again, how were these judgments made and who made them? Was it the health-care provider for the control group and the experimenter for the experimental group? What was the specific question asked to patients, was it standardized, and why was a 10-point (as opposed to, for example 5-point) scale used: These are examples from the Measurements section that need to be discussed more clearly and in more detail.
2. There should be some discussion as to why it was hypothesized that blood pressure and heart-rate would not be influenced by the therapy dog (lines 48-49) but the Barthel index would be? Indeed, other studies, and several mentioned in the Discussion section (line 223) have found an effect of AAT on heart-rate and blood-pressure. Also, in the discussion of the studies that found a positive association (lines 223-228) it would be helpful to briefly explain in what settings/contexts and with what populations such findings were seen.
3. In the Results section all t-test values should be included (e.g. t-value, df value), not just the p-values (e.g. lines 169,170, 185, etc.). The vertical axis labels of the figures seem more appropriate as figure titles; it would be helpful to instead have labels related to the dependent variables shown. Clarification is needed for the description of Figure 2 in lines 190-192. It is especially unclear because the text in lines 185-188 is also ambiguous – perhaps restate the analysis done, the exact question in focus, or explicitly which results are meant by “a comparison of results for the experimental group…”(line 185).
4. It is confusing to have all the citations placed at the end of a sentence when the sentence contains multiple statements that need citations. For example: lines 20 & 21 include the two separate statements, 1) that “a large number of hospitalization patients have pets” and 2) that contact with an animal might beneficially remind them of home – is the Fine (2006) citation related to both of these points? If not, a second citation would be needed. If so, the sentence could be re-worded to be more clear. Another example are the findings and citations reported on lines 35-37. And, a final illustrative example can be found on lines 266-268. Which studies link to the findings that an animal can distract from attention to pain, and which for reflection? It is also unclear from the current wording what is meant here by “reflection on the participants”. Also, the studies mentioned on line 28 need citations, as do the claims on lines 32 and 33-34.
5. A thorough edit for wording and clarity will improve this paper. For example, “the older people” (line 5) would read more clearly as “elderly” and “this strategy” (line 19) should be “these strategies”. Line 204 should say “paired-samples t-test” instead of “Pair T-Test”. Particularly confusing wording includes lines 63, 212, 213, 325 (what is meant by “optional frequency”?), and 245 (what is meant by “better oriented”?). Also, the capitalization of journal titles is inconsistent in the references.
Author Response
Dear reviewer,
Thank you very much for your suggestions. We have tried to fix everything and we think that it was very helpfull for the article.
Best regards
Kristýna Machová
Comment | Authors response |
1. A main study variable was patient mood or ‘sense of satisfaction’ (line 76) but with the current wording it is unclear exactly how this was measured. First, with the current wording “A mood was evaluated by the Likert scale to assess the sense of satisfaction of the patient” (line 76) it is unclear if the authors were trying to asses patients’ general moods or their sense of satisfaction - or if these are considered the same construct or different. The rest of the sentence (lines 76-77)…”which made it possible to ascertain their current attitude and the approximate degree of the same” is one example of wording whose meaning is impossible to decipher. Most importantly, it is unclear how exactly patients’ sense of satisfaction was actually evaluated – if they rated their satisfaction on a scale of 1-10, or if this rating was judged based on a more general answer, and if so by whom? The authors do write on line 80: “Their answers were judged on a scale of one to ten”. But again, how were these judgments made and who made them? Was it the health-care provider for the control group and the experimenter for the experimental group? What was the specific question asked to patients, was it standardized, and why was a 10-point (as opposed to, for example 5-point) scale used: These are examples from the Measurements section that need to be discussed more clearly and in more detail. | 1) We change the terminology at general mood of patients, which is proper term 2) A mood was evaluated by a Likert scale that allowed patients to evaluate their general mood on a scale of 1 to 10. [10-11] on every day of therapy. The people in the experimental and control groups were asked by the same therapist to evaluate how they felt. 3) They were asked to rate by themself their mood from 1 to 10 4) They instruction was: Rate your mood you have right now on a scale of 1 to 10, with 1 being the worst mood and 10 being the best mood. The 10-point scale was chosen for better understanding reported by patients compared to a 5-point scale, that patients confused with grading at school, where 1 is the best classification. |
There should be some discussion as to why it was hypothesized that blood pressure and heart-rate would not be influenced by the therapy dog (lines 48-49) but the Barthel index would be? Indeed, other studies, and several mentioned in the Discussion section (line 223) have found an effect of AAT on heart-rate and blood-pressure. Also, in the discussion of the studies that found a positive association (lines 223-228) it would be helpful to briefly explain in what settings/contexts and with what populations such findings were seen. | 5) Because their levels of blood pressure and heart rate weren´t at the beginning pathological. · To this end, the hypothesis was that a positive influence would be exerted, pertaining to the general mood and Barthel index of the patient, thanks to their activation and mediating pleasant experiences · We expect the blood pressure and heart rate to be the same in both groups. The value will remain at the physiological level. Therapy will not lead to any pathological increase or decrease. 6) All the studies were working with patients or participants with pathological levels of blood pressure, cardiovascular disease or under acute stress |
In the Results section all t-test values should be included (e.g. t-value, df value), not just the p-values (e.g. lines 169,170, 185, etc.). The vertical axis labels of the figures seem more appropriate as figure titles; it would be helpful to instead have labels related to the dependent variables shown. Clarification is needed for the description of Figure 2 in lines 190-192. It is especially unclear because the text in lines 185-188 is also ambiguous – perhaps restate the analysis done, the exact question in focus, or explicitly which results are meant by “a comparison of results for the experimental group…”(line 185). | 7) The experimental group demonstrated a statistically significant difference in shift in mood prior to and following a therapy session with the dog (the value of the Wilcoxon pair test criterion W = 22; df = 22; p < 0,001). However, variance between satisfaction in the morning and the second measurement was also seen in the control group (the value of the Wilcoxon pair test criterion W = 47,5; df = 24; p<0,005). Complete results are shown in Figure 1.
8) Figures were corrected |
It is confusing to have all the citations placed at the end of a sentence when the sentence contains multiple statements that need citations. For example: lines 20 & 21 include the two separate statements, 1) that “a large number of hospitalization patients have pets” and 2) that contact with an animal might beneficially remind them of home – is the Fine (2006) citation related to both of these points? If not, a second citation would be needed. If so, the sentence could be re-worded to be more clear. Another example are the findings and citations reported on lines 35-37. And, a final illustrative example can be found on lines 266-268. Which studies link to the findings that an animal can distract from attention to pain, and which for reflection? It is also unclear from the current wording what is meant here by “reflection on the participants”. Also, the studies mentioned on line 28 need citations, as do the claims on lines 32 and 33-34. | 9) corrected |
A thorough edit for wording and clarity will improve this paper. For example, “the older people” (line 5) would read more clearly as “elderly” and “this strategy” (line 19) should be “these strategies”. Line 204 should say “paired-samples t-test” instead of “Pair T-Test”. Particularly confusing wording includes lines 63, 212, 213, 325 (what is meant by “optional frequency”?), and 245 (what is meant by “better oriented”?). Also, the capitalization of journal titles is inconsistent in the references. | 10) corrected |

Reviewer 2 Report
I enjoyed reading about this fascinating study into the effectiveness of AAT for long-term older patients in hospitals. The paper is generally well written and I have just a few comments/suggestions for improvement:
Referencing seems quite sparse in this section - generally just one per paragraph. Given that this is the section that places the study in context and provides rationale and underpinning insights, I think this needs to be addressed. There is a wide body of literature on many of these issues that could be engaged with to provide better support for the claims made and to direct readers to further relevant studies.
What kind of illnesses did the patients have? Were they all physical, or also some cognitive? Did many participants have dementia? I think this is important to know, and if there are any differences between groups in terms of how they reacted to the dog
Writing - the paper would benefit from proof reading by a native speaker for the first part. Whilst it is generally fine, there are some places where the structure or phrasing is a little unclear and this disrupts the flow of the argument.
Author Response
Dear reviewer,
Thank you very much for your suggestions. We have tried to fix everything and we think that it was very helpful for the article.
Best regards
Kristýna Machová
Comment | Authors response |
Referencing seems quite sparse in this section - generally just one per paragraph. Given that this is the section that places the study in context and provides rationale and underpinning insights, I think this needs to be addressed. There is a wide body of literature on many of these issues that could be engaged with to provide better support for the claims made and to direct readers to further relevant studies. | 1) We have improved the more detailed layout of the studies to better refer to the information. We have tried to reduce references a little, so that the list of literature is not too extensive because some journals prefer sparse references. However, if you find it more appropriate to extend the list after making your changes, we will add appropriate references. |
What kind of illnesses did the patients have? Were they all physical, or also some cognitive? Did many participants have dementia? I think this is important to know, and if there are any differences between groups in terms of how they reacted to the dog | 2) The disease that affected patients was stroke, mild dementia or mild cognitive disease, or cancer. |
Writing - the paper would benefit from a proof reading by a native speaker for the first part. Whilst it is fine, there are some places where the structure or phrasing is unclear and the flow of the argument. | 3) The article was checked by native speaker. |
